# Durable superhydrophobic coatings for prevention of rain attenuation of 5G/weather radomes

Jinfei Wei [1], Jiaojiao Zhang [1], Xiaojun Cao [2], Jinhui Huo [2], Xiaopeng Huang [1] & Junping Zhang [1,3] ✉

Superhydrophobic coatings are expected to solve the rain attenuation issue of 5G radomes. However, it is very challenging to design and construct such superhydrophobic coatings with good impalement resistance, mechanical robustness, and weather resistance, which remains as one of the main bottlenecks hindering their practical applications. Here, we report the design of superhydrophobic coatings with all these merits mentioned above by spray-coating a suspension of adhesive/fluorinated silica core/shell microspheres onto substrates. The core/shell microspheres are formed by phase separation of the adhesive and adhesion between the adhesive and fluorinated silica nanoparticles. The coatings have an approximately isotropic three-tier hierarchical micro-/micro-/nanostructure, a dense but rough surface at the nanoscale, and chemically inert composition with low surface energy. Consequently, the coatings show excellent impalement resistance, mechanical robustness and weather resistance compared with previous studies, and the mechanisms are revealed. Furthermore, we realize large-scale preparation, extension, and practical application of the coatings for efficiently preventing rain attenuation of 5G/weather radomes. By taking these advantages, we believe that the superhydrophobic coatings have great application potential and market prospect. The findings here will boost preparation and real-world applications of superhydrophobic coatings.

The 5th Generation (5G) mobile communication technology has become an important infrastructure supporting digital, networked and intelligent transformation of our society, due to its advantages of high data rate, low delay and multiple data transfer paths, etc[1]. Thus, numerous 5G base stations have been built, e.g., 2.3 million at the end of 2022 in China[2]. 5G radome is an important part of 5G base stations for protecting the antenna system from interference of complex outdoor environment, improving accuracy and reliability, and extending service life, etc. However, during rainfall 5G signals can be easily disturbed by raindrops or water films on the radomes[3,4]. This is because water with high dielectric constant (~80 at 25 °C) absorbs and reflects large amounts of electromagnetic wave, resulting in serious attenuation of 5G signals[5,6]. The rain attenuation issue of 5G radomes is one of the bottlenecks restricting 5G communication development. In fact, rain attenuation is a common drawback of radomes used for 5G base stations, weather radars and military radars, etc.

Biomimetic superhydrophobic coatings with high water contact angle (CA > 150°) and low sliding angle (SA < 10°) have broad application prospects in anti-adhesion of liquids, inhibiting liquid spreading and self-cleaning, etc[7–13]. Thus, superhydrophobic coatings are

[1]Center of Eco-Material and Green Chemistry, Lanzhou Institute of Chemical Physics, Chinese Academy of Sciences, 730000 Lanzhou, PR China. [2]Shandong Xinna Superhydrophobic New Materials Co. Ltd., 265402 Yantai, PR China. [3]Center of Materials Science and Optoelectronics Engineering, University of Chinese Academy of Sciences, 100049 Beijing, PR China. ✉e-mail: jpzhang@licp.cas.cn

expected to solve the rain attenuation issue of radomes. Nevertheless, radomes are usually used in outdoor conditions. The radomes are persistently exposed to the complex outdoor environment for a long time, such as continuous raindrops impact, sand and dust erosion, ultraviolet radiation, corrosive media, high/low temperature, icing/melting and snow. Therefore, ideal superhydrophobic coatings used for prevention of rain attenuation of 5G/weather radomes should fulfill the advantages of excellent impalement resistance, mechanical robustness, and weather resistance. However, all these three key factors are very challenging, hindering practical applications of superhydrophobic coatings[14–16].

A number of superhydrophobic coatings with good impalement resistance[17,18], mechanical robustness[13,19–23] or weather resistance[24,25] have been reported so far. (1) Re-entrant structures[17,26] and multi-tier hierarchical structures[18,27] are helpful to achieve good impalement resistance, but such structures with high aspect ratio can be easily destroyed by abrasion due to mechanical stress concentration[15,21,28]. Elastic materials like fluorinated epoxy resin with perfluoropolyether can also improve dynamic superhydrophobicity[29], but are too expensive to be used on a large scale. (2) Micro-skeleton protection[19,20,30], self-healing[31,32], self-similar structures[33,34] and adhesives[22,35–37] are good strategies to enhance mechanical robustness. Among them, the use of adhesives is convenient for large-scale preparation[19,22]. However, adhesives often significantly increase surface energy of the coatings by covering low-surface-energy components and thus cause poor (dynamic) superhydrophobicity[25,38]. We recently reported an adhesive-based mechanically robust superamphiphobic coating for anti-icing, but neglected its impalement resistance and weather resistance[19]. As a result, the outdoor service life of the coating is not good enough. Additionally, the coating suspension containing ammonia is very pungent, which makes it very inconvenient for large-scale spray-coating. (3) Chemically inert inorganics[39,40] and polymers (e.g., silicone resin and rubber)[41–43] can enhance weather resistance, but their hydrophilicity often results in poor (dynamic) superhydrophobicity. Now, it is still very difficult to simultaneously integrate these multiple merits in one superhydrophobic coating, especially by using simple materials and methods[16,23]. Unfortunately, however, simple methods with commercially available materials are prerequisites for large-scale preparation and practical applications of superhydrophobic coatings.

Here, we report application-oriented design of simultaneously impalement-resistant, mechanically robust, and weather-resistant superhydrophobic coatings by a simple method with common materials. Based on theoretical analysis and previous studies, the design criteria of the coatings are proposed. Then, the coatings are prepared by spray-coating a suspension of adhesive/fluorinated silica core/shell microspheres onto substrates. The coatings show excellent impalement resistance, mechanical robustness and weather resistance as verified by various tests. Also, we realize large-scale preparation, extension, and practical application of the coatings for efficiently preventing rain attenuation of 5G/weather radomes.

## Results and discussion
### Design of durable superhydrophobic coatings
According to the Supplementary Note, to enhance dynamic superhydrophobicity, we designed a suspension through formation of polyolefin adhesive (POA) microspheres via phase separation followed by wrapping the microspheres with the synthesized perfluorodecyl polysiloxane modified silica (fluoroPOS@silica) nanoparticles via adhesion between them (Fig. 1a, b). This strategy (i) avoids embedding of the fluoroPOS@silica nanoparticles in POA and hence maintains low surface energy; and ii) forms POA/fluoroPOS@silica with core/shell structure, i.e., two-tier hierarchical micro-/nanostructure. Moreover, by faster evaporation of the non-solvent (ethanol) than the solvent (butyl acetate) during spray-coating, the POA microspheres partly dissolve and link with each other, which causes aggregation of the POA

microspheres. Thus, the POA/fluoroPOS@silica coating has a three-tier hierarchical micro-/micro-/nanostructure. Furthermore, the fluoroPOS@silica nanoparticles densely and firmly pack on the POA microspheres owing to evaporation of solvents and adhesion between them, which forms a very dense but rough surface at the nanoscale, i.e., very small $d$ (the mean distance between protrusions) in Supplementary Formula (5).

To achieve high mechanical robustness, hydrophobic POA ($CA_{water}$ = 95.5°) with Shore A hardness 55 was selected to strengthen the linkages among the particles in the coating and to form strong adhesion between the coating and the substrate. Such an adhesive can enhance mechanical robustness without sacrificing superhydrophobicity. Also, the coating composed of POA/fluoroPOS@silica microspheres and their aggregates has an approximately isotropic micro-/micro-/nanostructure. Once the surface of the coating is damaged, the exposed new surface still has almost the same performance as the original surface.

To achieve high weather resistance, we designed the coating using chemically inert components including fluoroPOS, silica nanoparticles and POA.

### Preparation of POA/fluoroPOS@silica coatings
According to the above design criteria, we prepared the POA/fluoroPOS@silica superhydrophobic coatings. First, the fluoroPOS@silica nanoparticles were prepared by hydrolytic co-condensation of *1H,1H,2H,2H*-perfluorodecyltriethoxysilane (PFDTES), tetraethoxysilane (TEOS) and sodium methylsilicate in the ethanol suspension of hydrophilic silica nanoparticles (10-20 nm) (Fig. 1a and Supplementary Fig. 1). A small amount of TEOS can enhance superamphiphobicity of the fluoroPOS@silica coating (Supplementary Fig. 2), as TEOS helps hydrolytic condensation of PFDTES on the surface of silica nanoparticles[44]. Obviously, fluoroPOS@silica nanoparticles with higher superamphiphobicity are helpful to improve superhydrophobicity of the POA/fluoroPOS@silica coatings. Note that sodium methylsilicate acts as both the reactant and the catalyst. This is substantially different from the conventional hydrolytic condensation of silanes with ammonia as the catalyst. Replacing ammonia with sodium methylsilicate solved the pungent smell issue of conventional coating suspensions containing ammonia. Then, the semi-solid fluoroPOS@silica nanoparticles containing ethanol was obtained by centrifugation. Subsequently, under vigorous stirring the fluoroPOS@silica nanoparticles were slowly added into the POA solution in butyl acetate, during which ethanol (non-solvent) induced gradual phase separation of POA (Fig. 1b). Finally, the POA/fluoroPOS@silica coatings were fabricated by spraying the homogeneous POA/fluoroPOS@silica suspension onto ABS plates and curing at room temperature for 24 h (Supplementary Fig. 3). The mass ratio of POA to fluoroPOS@silica nanoparticles has great influences on performance of the coatings including impalement resistance and mechanical robustness (Supplementary Fig. 4 and Supplementary Discussion). The mass ratio of POA to fluoroPOS@silica nanoparticles is 0.4:1 and the coating thickness is ~100 μm unless otherwise specified.

According to the SEM image at low magnification (Fig. 1c), the coating has a two-tier hierarchical micro-/microstructure composed of POA/fluoroPOS@silica microspheres (3–5 μm) and their micro-aggregates (10–30 μm). In the spray-coating process, ethanol evaporates faster than butyl acetate, which induces partly dissolution and linking among the POA microspheres and thus forms the micro-aggregates. Solvent evaporation and interparticle adhesion result in densely packing of fluoroPOS@silica nanoparticles on the surface of POA microspheres (Fig. 1d). Therefore, the coating has a three-tier hierarchical micro-/micro-/nanostructure, which helps enhance dynamic superhydrophobicity[27,45]. The cross-sectional SEM image shows that the coating has an approximately isotropic micro-/micro-/nanostructure (Fig. 1e). The energy dispersive spectrum and elemental mapping demonstrate that the surface and cross-section of the coating

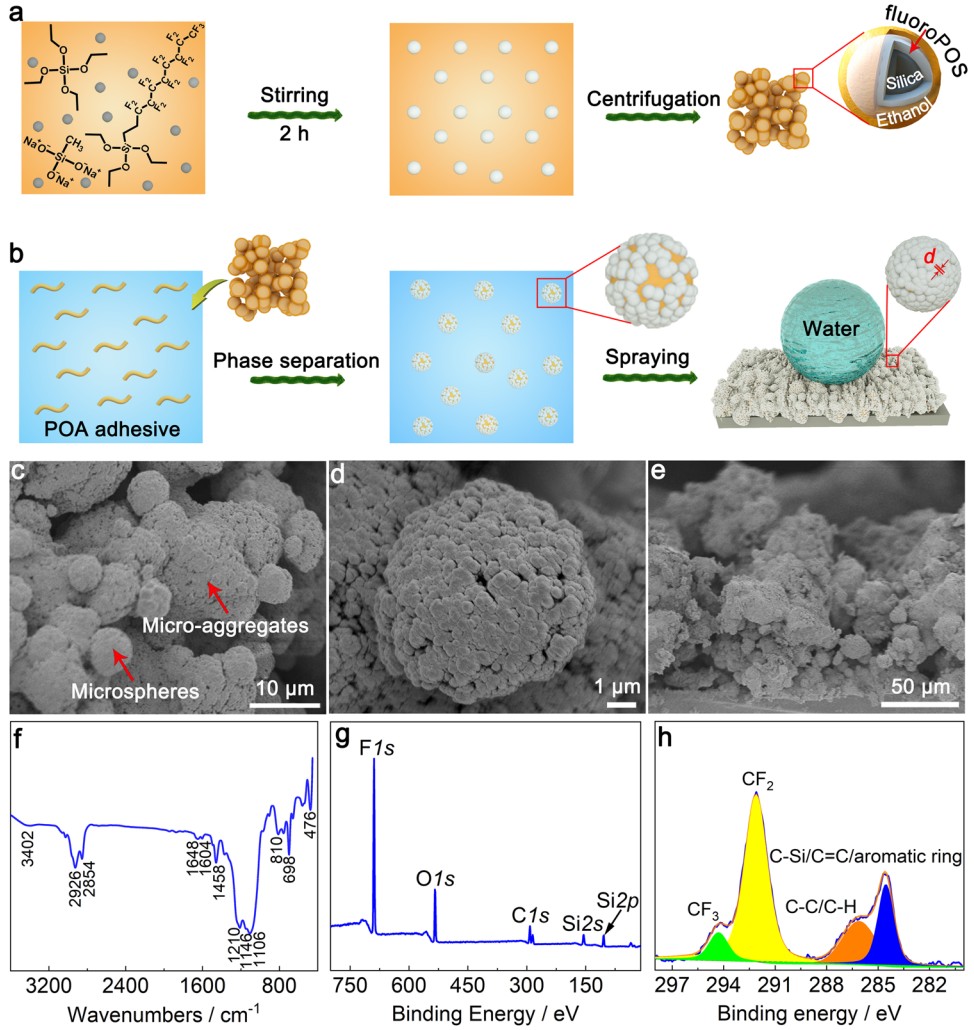

**Fig. 1 | Preparation and characterization of the POA/fluoroPOS@silica coatings.** Schematic preparation of the (**a**) fluoroPOS@silica nanoparticles and **b** POA/fluoroPOS@silica coatings. "*d*" in (**b**) is the mean distance between protrusions. **c, d** Surface and (**e**) cross-sectional scanning electron microscopic (SEM) images of the coating. **f** Fourier transform infrared (FTIR) spectrum of POA/fluoroPOS@silica. **g** X-ray photoelectron spectrum (XPS) and (**h**) high-resolution C 1s spectrum of the POA/fluoroPOS@silica coating.

are chemically uniform (Supplementary Figs. 5 and 6). Such an isotropic micro-/micro-/nanostructure with uniform chemical composition helps enhance mechanical robustness[34].

From the FTIR spectrum of POA/fluoroPOS@silica, the characteristic peaks of all components can be seen (Fig. 1f). The weak peak at 3402 cm$^{-1}$ is assigned to the residual -OH[46]. The peaks of Si-O-Si (1106 and 810 cm$^{-1}$) and Si-O (476 cm$^{-1}$) appear[47,48]. The strong peaks at 1210 and 1146 cm$^{-1}$ are assigned to C-F and silsesquioxane, respectively[49]. In addition, the peaks corresponding to C=C (1648 cm$^{-1}$) and aromatic ring (1604 cm$^{-1}$) of POA appear[50,51].

The surface chemical composition of the POA/fluoroPOS@silica coating was studied by XPS (Fig. 1g, h). The C 1s (292.09 eV), O 1s (533.66 eV), F 1s (689.28 eV), and Si 2p (104.47 eV) peaks are detected. The F/C/O/Si atomic ratio is 2.32/1.50/1.48/1 and the F content is up to 36.81 at% (Supplementary Table 1). Thus, there are numerous -Si(CH$_2$)$_2$(CF$_2$)$_7$CF$_3$ groups on the surface of the coating to reduce its surface energy[47]. The C 1s peak is assigned to CF$_3$ (294.3 eV), CF$_2$ (292.1 eV), C-C/C-H (286.1 eV) and C-Si/C=C/aromatic ring (284.5 eV)[35,51,52].

## Static and dynamic superhydrophobicity

The POA/fluoroPOS@silica coating shows excellent static superhydrophobicity. Water droplets are spherical on the coating (CA =

165.1°) and can easily roll off the slightly inclined coating (1.0°, Fig. 2a). In water, strong light reflection happens on the surface of the coating (Fig. 2b). These results prove existence of an air cushion at the coating–water interface, i.e., water is in the Cassie-Baxter state[53,54].

The POA/fluoroPOS@silica coating also shows excellent dynamic superhydrophobicity, which is very important for practical applications. A water jet can bounce off the coating without leaving any trace (Supplementary Fig. 7). A water droplet (10 μL) released from 1 cm height can bounce 8 times on the coating (Supplementary Movie 1). The solid-liquid contact time and bounce height in the first impact/bounce cycle are 13 ms and 3.7 mm, respectively (Fig. 2c). Moreover, the coating keeps dry without any change in superhydrophobicity during 120 s high-pressure water jetting at 50 kPa (Fig. 2d, Supplementary Fig. 8 and Supplementary Movie 2). Also, during 20 h of the simulated rainfall test, the coating remains dry and maintains the original superhydrophobicity (Fig. 2e and Supplementary Movie 2). The excellent dynamic superhydrophobicity is attributed to its three-tier hierarchical micro-/micro-/nanostructure[27], dense but rough surface at the nanoscale[45] and low surface energy[19].

The POA/fluoroPOS@silica coating has good anti-fouling and self-cleaning properties (Supplementary Fig. 9). After contacting with sewage droplets or being immersed in sewage, the blank ABS plate was seriously contaminated, whereas the coated ABS plate kept clean.

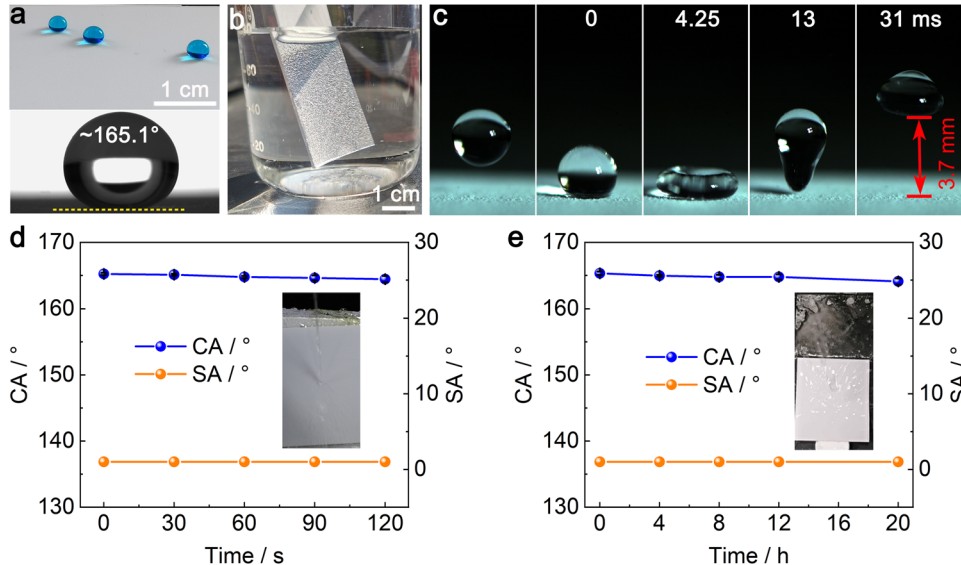

**Fig. 2 | Static and dynamic superhydrophobicity of the POA/fluoroPOS@silica coating.** Photographs of the POA/fluoroPOS@silica coating (**a**) with water droplets (CA ~ 165.1°), (**b**) in water and (**c**) with impact/bounce of a water droplet. Changes in CA and SA of the coating during the (**d**) high-pressure water jetting test and (**e**) simulated rainfall test. The insets show the test processes. Data in (**d**) and (**e**) are shown as mean ± SD, $n = 5$.

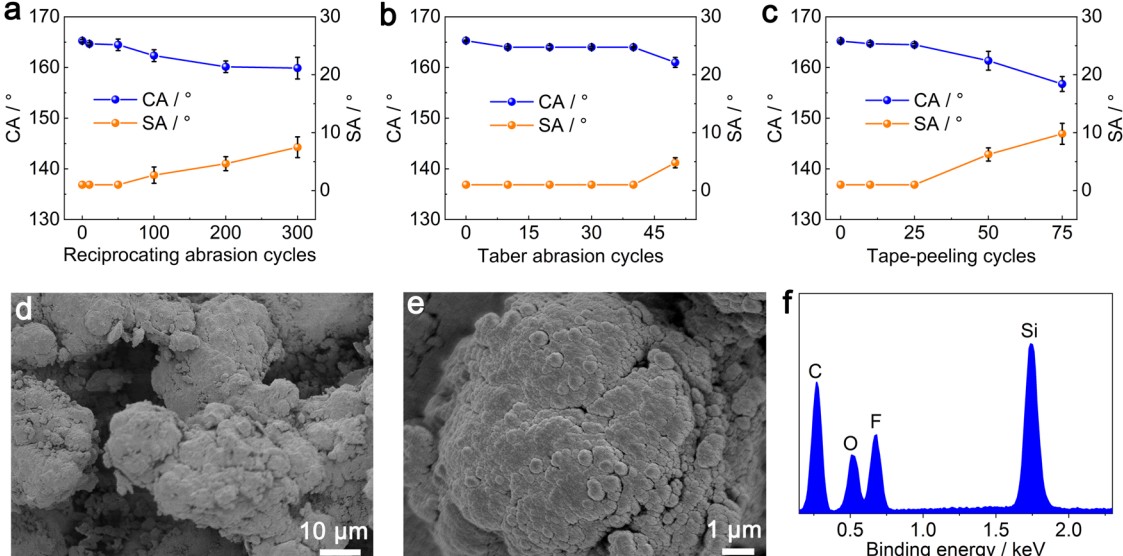

**Fig. 3 | Mechanical robustness of the POA/fluoroPOS@silica coating.** Changes in CA and SA of the POA/fluoroPOS@silica coating during the (**a**) reciprocating abrasion, (**b**) Taber abrasion and (**c**) tape-peeling tests. **d, e** SEM images and (**f**) EDS spectrum of the coating after 300 cycles reciprocating abrasion. Data in (**a**–**c**) are shown as mean ± SD, $n = 5$.

Additionally, water droplets can efficiently take away sand particles on the coated ABS plate.

## Mechanical robustness

The POA/fluoroPOS@silica coating has excellent mechanical robustness according to the reciprocating sandpaper abrasion test (2.3 kPa load, 1000 meshes), the Taber abrasion test (125 g load, CS 10 grinding wheel, ASTM D4060), and the tape-peeling test (2.3 kPa load, 3 M tape, ASTM D3359). In the reciprocating abrasion test, the original superhydrophobicity was maintained in the first 50 cycles, and then slightly declined (Fig. 3a). Even after 300 cycles (120 m), the coating still showed good superhydrophobicity (CA = 159.9°, SA = 7.5°). In the Taber abrasion test, the coating kept the original superhydrophobicity in the first 40 cycles, and still showed good superhydrophobicity after 50 cycles (CA = 161.0°, SA = 4.8°, Fig. 3b). In the tape-peeling test, the

coating kept the original superhydrophobicity in the first 25 cycles, and then gradually declined in the subsequent 50 cycles to a CA of 156.7° and a SA of 9.8° (Fig. 3c). After 300 cycles reciprocating abrasion, 50 cycles Taber abrasion or 75 cycles tape-peeling, water droplets were still spherical on the coating and could roll off without leaving any trace (Supplementary Fig. 10 and Supplementary Movie 3). Also, such slight increase in the SA would not result in pinning of water droplets under normal impact conditions like impacting from 1 cm height with a speed of 0.44 m/s or even from 100 cm height with a speed of 4.43 m/s (Supplementary Figs. 11, 12 and Supplementary Movie 4). Moreover, the coating remained intact after the hundred-grid adhesion strength test (Supplementary Fig. 13, ASTM D3359), which indicates good adhesion strength between the coating and the ABS plate. In addition, the coating could withstand intensive hand rub under water and abrasion with steel wool (Supplementary Movie 5). In

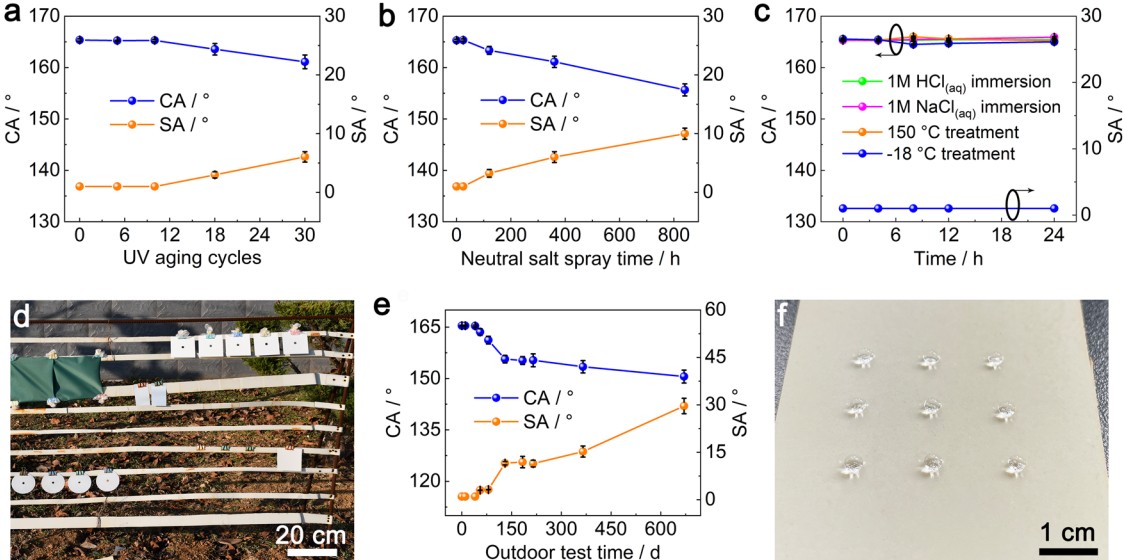

**Fig. 4 | Weather resistance of the POA/fluoroPOS@silica coating.** Changes in CA and SA of the POA/fluoroPOS@silica coating during (**a**) the UV aging test, (**b**) the neutral salt spray test, (**c**) immersion in corrosive liquids or treatment at harsh temperature. **d** Long-term weather resistance test of the coating in the outdoor environment in Yantai, Shandong, PR China, (**e**) changes in CA and SA during the test and (**f**) photograph of the coating with water droplets after 670 d of the test. Data in (**a**–**c**) and (**e**) are shown as mean ± SD, $n = 5$.

the process of hand rub under water, the coating remained dry without any change in superhydrophobicity. The mechanical robustness of the coating is superior to most reported superhydrophobic coatings, but is still not as good as the state-of-the art ones (Supplementary Table 2)[20,22]. Nevertheless, our one-layer spray-coating method is pretty simple. Note that we must give consideration simultaneously to impalement resistance, mechanical robustness, and weather resistance rather than merely mechanical robustness to move superhydrophobic coatings to real-world applications.

The excellent mechanical robustness is attributed to the following mechanism. The coating has an approximately isotropic micro-/micro-/nanostructure with uniform chemical composition (Fig. 1c–e and Supplementary Fig. 6). With increasing the abrasion or peeling cycles, the newly exposed surface is still almost the same as the original one (Fig. 3d–f, Supplementary Fig. 14, and Supplementary Table 3). Therefore, the mechanical robustness of the coating is closely related to its thickness (Supplementary Fig. 15). With increasing the thickness from 50 μm to 200 μm, the coatings can withstand more Taber abrasion cycles. Also, POA can strengthen the linkages among the particles in the coating and form strong adhesion between the coating and the substrate[55,56].

## Weather resistance

The POA/fluoroPOS@silica coating has excellent weather resistance, which ensures its practical applications in the outdoor conditions. During 30 cycles of the UV aging test with 4 h UV irradiation at 60 °C and 4 h condensation at 50 °C in each cycle (GB/T 14522-2008), the superhydrophobicity remained constant in the first 10 cycles, and then slightly declined to a CA of 161.1° and a SA of 6.0° (Fig. 4a). After 840 h of the neutral salt spray corrosion test (5 wt% NaCl, 35 °C, ASTM B117), the superhydrophobicity only slightly decreased to a CA of 155.6° and a SA of 10.0° (Fig. 4b). There was no change in appearance of the coating after both tests (Supplementary Fig. 16). Additionally, the coating could maintain the original superhydrophobicity after immersion in corrosive liquids (1 M HCl$_{(aq)}$ and 1 M NaCl$_{(aq)}$) for 24 h or treatment at 150 °C/−18 °C in the air for 24 h (Fig. 4c), which means excellent corrosion resistance and temperature stability. Note that the SA of the coating was 1° during all the four tests in Fig. 4c.

The POA/fluoroPOS@silica coating has good anti-icing performance (Supplementary Fig. 17 and Supplementary Movie 6). At −20 °C and 97% relative humidity, the water droplets (60 μL) completely froze after 168.7 ± 4.1 s on the ABS plate but delayed to 301.7 ± 10.3 s on the POA/fluoroPOS@silica coated ABS plate. Also, the coating could evidently reduce ice adhesion strength from 121 ± 9.3 kPa (the ABS plate) to 39.2 ± 2.8 kPa. The good anti-icing performance is originated from the stable air cushion at the coating–water interface, which greatly reduces the contact area and inhibits heat transfer between them, and also weakens the ice adhesion strength on the coating[19,22,57].

The POA/fluoroPOS@silica coating also shows excellent long-term weather resistance in the outdoor environment (Fig. 4d). The weather conditions are shown in Supplementary Table 4. Long-term outdoor tests of superhydrophobic coatings are rare in the literature, but very important for their practical applications. In the outdoor test, the original superhydrophobicity was maintained in the first 40 d, and then gradually declined (Fig. 4e). Even after 670 d, the coating was still superhydrophobic (CA = 150.6°, SA = 29.6°) without evident change in the appearance (Fig. 4f and Supplementary Fig. 18). The long-term outdoor weather resistance of the coating is superior to the reported superhydrophobic coatings (Supplementary Table 5)[39,41–43,58–61].

The excellent weather resistance, corrosion resistance and temperature stability are mainly because the coating is composed of chemically inert components.

## Large-scale preparation and extension

Large-scale preparation is the prerequisite for practical applications of the POA/fluoroPOS@silica coating in various fields such as preventing rain attenuation of radomes. As the materials and method are very simple, we have realized large-scale preparation of the POA/fluoroPOS@silica suspension, ~500 L per day with a low cost of ~17.3 USD/L (Fig. 5a and Supplementary Fig. 19). Also, we have achieved large-scale preparation of the POA/fluoroPOS@silica coating up to thousands of square meters (~5 m²/L), as the coating can be prepared via simple one-step spray-coating at room temperature. The large coatings have the same dynamic superhydrophobicity and mechanical stability as those prepared at the laboratory scale (Fig. 5b–d). In addition, the POA/fluoroPOS@silica coating outperforms some representative

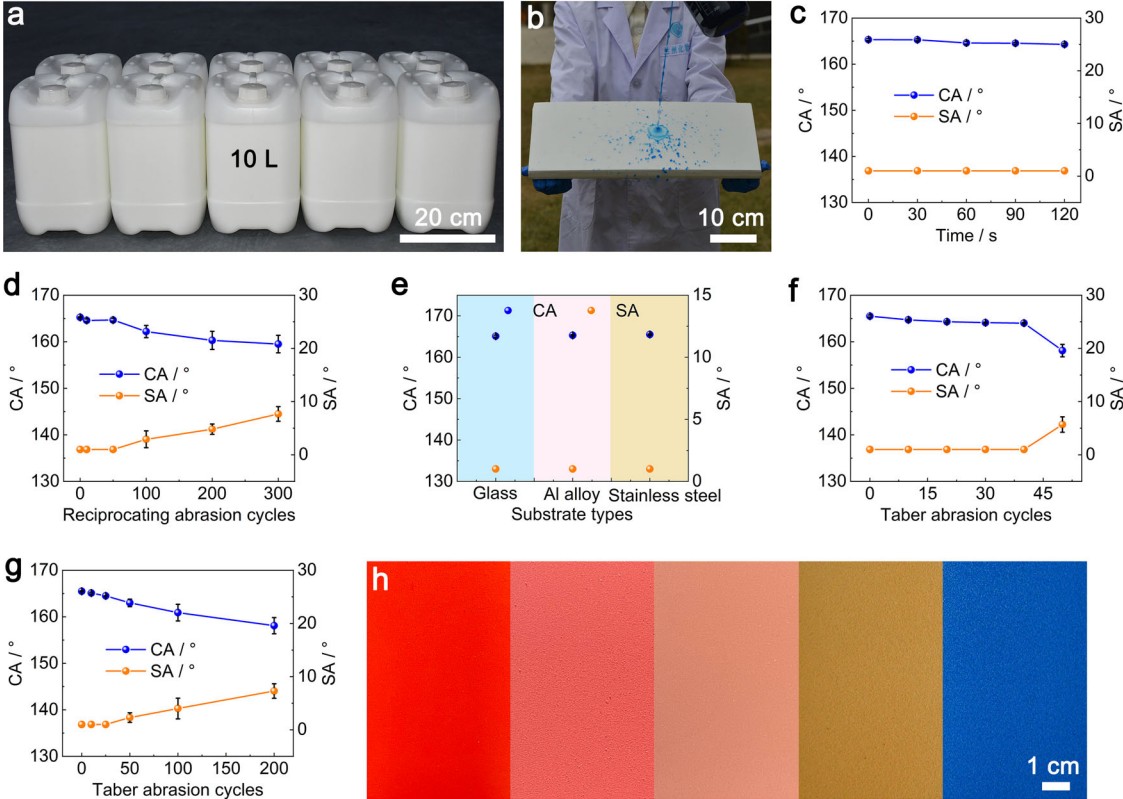

**Fig. 5 | Large-scale preparation and extension of POA/fluoroPOS@silica coatings.** Large-scale preparation of the (**a**) POA/fluoroPOS@silica suspension and (**b**) POA/fluoroPOS@silica coating. Changes in CA and SA of the large coating during the (**c**) high-pressure water jetting test and (**d**) reciprocating abrasion test. **e** Superhydrophobicity of the coatings on various substrates. Changes in CA and SA of the coatings (**f**) on the Al alloy and (**g**) with silicone-modified polyester as the primer during the Taber abrasion test. **h** Photographs of the colorful coatings. Data in (**c**–**g**) are shown as mean ± SD, $n = 5$.

commercial superhydrophobic products in terms of superhydrophobicity, mechanical robustness and simplicity (Supplementary Table 6).

Besides ABS plates, the POA/fluoroPOS@silica coating is applicable onto the other substrates including glass, Al alloy and stainless steel. All the coatings have the same superhydrophobicity and mechanical stability as that on the ABS plate (Fig. 5e, f and Supplementary Fig. 20). In addition, mechanical stability of the coating can be further improved by introducing a primer. For example, with silicone-modified polyester as the primer, the coating showed good superhydrophobicity even after 200 cycles Taber abrasion (CA = 158.1°, SA = 7.3°, Fig. 5g). Note that one-layer coating is preferred for practical applications, as multi-layer design consumes much more manpower and time. Also, colorful coatings can be prepared simply by adding proper pigment pastes in the POA/fluoroPOS@silica suspension (Fig. 5h and Supplementary Movie 7).

### Preventing rain attenuation of 5G/weather radomes

The application of the POA/fluoroPOS@silica coating for preventing rain attenuation of 5G radome was investigated by testing signals of four 5G millimeter-wave frequency bands (N257, N258, N260, and N261) using a custom-made setup with good repeatability for signal transmission (Supplementary Figs. 21 and 22). The low, middle, and high frequency points of each frequency band were tested. The coating has no influence on the signals (Fig. 6a, b and Supplementary Fig. 23), due to high dielectric strength of the coating (38.2 ± 0.88 kV/mm). For the blank radome, after 48 h simulated rainfall the signals of all frequency bands showed significant loss of −3.71--5.17 dB. In contrast, for the coated radome there was almost no signal loss in all frequency bands (−0.05--0.28 dB), which proves successful prevention

of rain attenuation of 5G radome by the coating. Additionally, the POA/fluoroPOS@silica coating outperforms those used in the field of antennas in terms of superhydrophobicity, stability and large-scale practical applications (Supplementary Table 7).

This is because the blank radome is hydrophilic (CA = 58.3°). Raindrops will adhere on its surface and form a water film during rainfall, resulting in serious signal loss (Fig. 6c, e). In contrast, the coated radome remains completely dry during rainfall owing to its excellent static and dynamic superhydrophobicity, and thus successfully prevents rain attenuation (Fig. 6d, e and Supplementary Movie 8).

Additionally, the coating can be applied on 5G radomes in different shapes (Fig. 6f, g and Supplementary Fig. 24). Also, the coating can solve the rain attenuation issue of weather radomes, which are greatly larger than 5G radomes (Fig. 6h and Supplementary Movie 9). We have achieved large-scale engineering application of the coating for preventing rain attenuation of 5G/weather radomes. Numerous 5G radomes and hundreds of weather radomes have been coated and used practically to solve the rain attenuation issue (Supplementary Fig. 25). After more than one year, the coated radomes still had good superhydrophobicity for preventing rain attenuation (Supplementary Movie 10), owing to excellent dynamic superhydrophobicity, mechanical robustness and weather resistance of the coating.

In summary, we proposed the design criteria of simultaneously impalement-resistant, mechanically robust, and weather-resistant superhydrophobic coatings, and then realized their preparation by a simple method with common materials. The advantages of the coatings are obvious compared with previous studies. This is because (i) the coatings have an approximately isotropic three-tier hierarchical micro-/micro-/nanostructure and a dense but rough surface at the nanoscale, and (ii) the coatings are composed of chemically inert

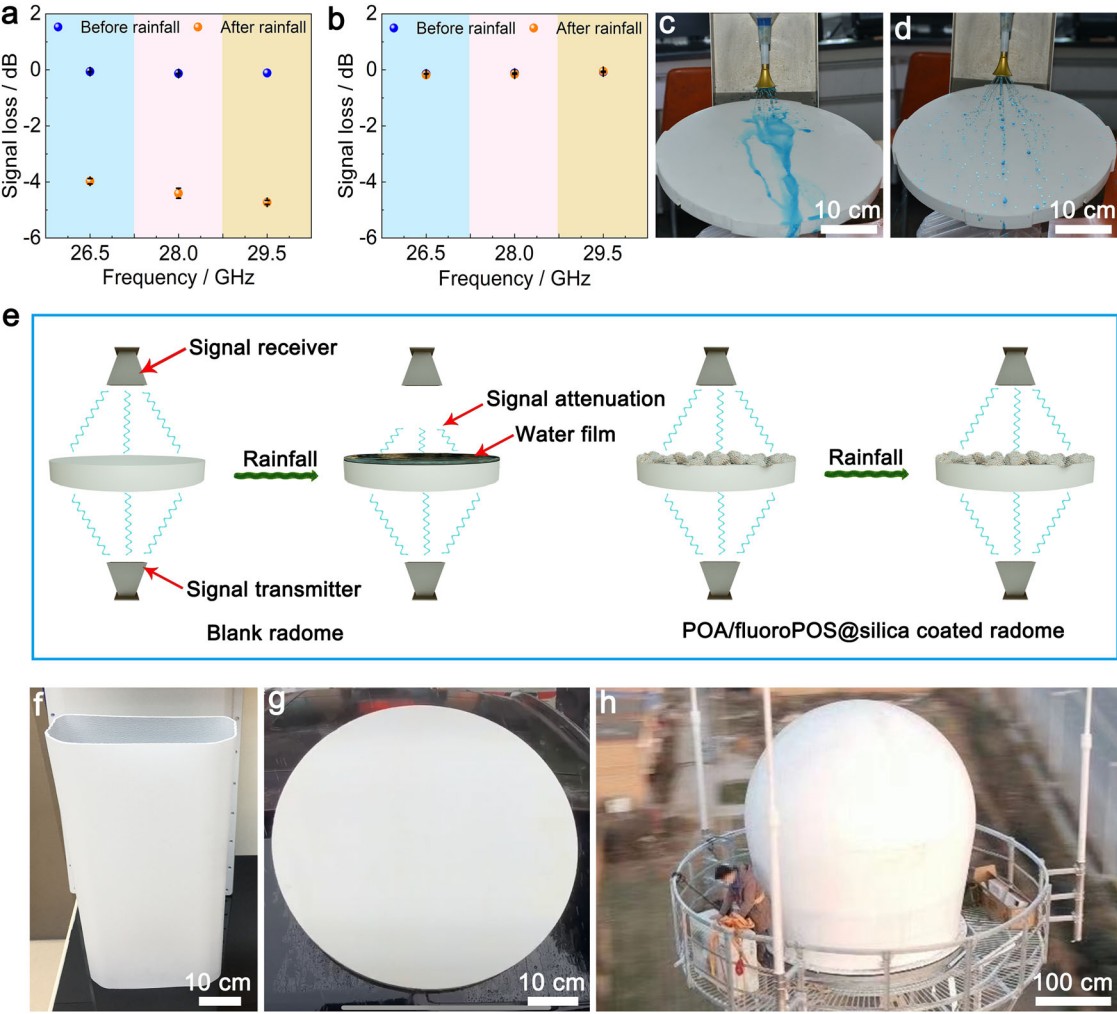

**Fig. 6 | Preventing rain attenuation of 5G/weather radomes by the POA/fluoroPOS@silica coating.** Signal loss of the N257 5G millimeter-wave frequency band through the (**a**) blank and (**b**) POA/fluoroPOS@silica coated 5G radomes before and after 48 h simulated rainfall. Photographs of simulated rainfall (colored with methylene blue) on the (**c**) blank and (**d**) coated 5G radomes. **e** Schematic mechanism of preventing rain attenuation of 5G radomes by the coating. Photographs of the (**f, g**) 5G radomes and (**h**) weather radome with the coating. Data in (**a**) and (**b**) are shown as mean ± SD, $n = 3$.

materials with low surface energy. Furthermore, we realized large-scale preparation, extension, and practical application of the coatings for efficient prevention of rain attenuation of 5G/weather radomes. This work is a precious example realizing design, preparation, large-scale production, and practical application of superhydrophobic coatings. Meanwhile, the impalement resistance, mechanical robustness, and weather resistance of the superhydrophobic coatings need to be further enhanced in order to have longer service life (e.g., 5–10 years) in practical applications, as discussed with our customers. We anticipate that this study will shed a light on the design and practical applications of superhydrophobicity coatings.

## Methods

### Synthesis of fluoroPOS@silica nanoparticles

First, 1.0 kg of hydrophilic silica nanoparticles (10–20 nm) were dispersed in 50 L of ethanol/water mixture by mechanical stirring (500 rpm, 10 min) and ultrasonication (5 L per batch, 10 min). Subsequently, 1.0 kg of sodium methylsilicate was added and continuously stirred for 10 min. Then, PFDTES (1.2 L) and TEOS (0.3 L) were added. After reaction at room conditions for 2 h under mechanical stirring, the fluoroPOS@silica suspension was formed, which was centrifugated to obtain the semi-solid fluoroPOS@silica nanoparticles containing ethanol.

### Fabrication of POA/fluoroPOS@silica superhydrophobic coatings

First, 1.4 kg of POA was dissolved in 5.5 kg of butyl acetate by mechanical stirring (500 rpm, 20 min). Subsequently, 3.5 kg of the fluoroPOS@silica nanoparticles were added slowly under stirring, during which phase separation of POA gradually occurred. After further stirring for 3 h, the uniform POA/fluoroPOS@silica suspension was obtained. Finally, the POA/fluoroPOS@silica coatings were prepared by spraying the POA/fluoroPOS@silica suspension (7.5 g) onto the ABS plates (200 cm²) using a spray gun (SATA 4400-110) at 0.2 MPa and curing at room temperature for 24 h. The coatings on the other substrates were prepared according to the same procedure.

The additional experimental information including materials, dynamic superhydrophobicity tests, mechanical robustness tests, weather resistance tests, evaluation of preventing rain attenuation performance, anti-icing performance tests and characterization is available in the Supplementary Information.

## Data availability

The data that support the findings of this study are available within the paper and Supplementary Information. Source data are provided with this paper.

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

## Acknowledgements

This work was supported by the National Natural Science Foundation of China (51873220 and 22275200, J.P.Z.), the Major Projects of the Science and Technology Plan of Gansu Province, China (21ZD4FA010, J.P.Z.), and the Key Program of the Lanzhou Institute of Chemical Physics, CAS (KJZLZD-4, J.P.Z.).

## Author contributions

Conceptualization: J.F.W., J.P.Z. Methodology: J.F.W., J.J.Z. Data Analysis: J.F.W. Investigation: J.F.W., J.J.Z., X.J.C., J.H.H., X.P.H. Supervision: J.P.Z. Writing—original draft: J.F.W., J.P.Z. Writing—review and editing: J.F.W., J.J.Z., X.J.C., J.H.H., X.P.H., J.P.Z.

## Competing interests

The authors declare no competing interests.
