## [Peer Review File · Nature Communications]

Reviewer comments first round

Reviewer #1 (Remarks to the Author):

This manuscript reports fabrication of superhydrophobic coatings via spray-coating a suspension that consists of silica particles, fluorinated compounds and adhesives. The authors study the mechanical stability of the coatings and also demonstrate application over a radome surface. The coatings exhibit certain level of mechanical robustness due to the hierarchical structures at multiple length scales together with the high thickness of the coating. Overall the novelty of the work is limited for publication in Nature Communications, considering that similar materials and methods have been studied extensively for fabricating superhydrophobic coatings. Please see below additional concerns.

- 1) "With increasing the abrasion or peeling cycles, the newly exposed surface is still the same as the original one". The thickness of the coating plays a key role in the mechanical robustness. The thickness of the coating was not even reported. The mechanical robustness should be studied as a function of the thickness.
- 2) There is significant increase in the sliding angle of the coatings after exposure to different effects (e.g. Figure 4e). These results limit the practical applicability.
- 3) There are issues regarding the appropriate usage of terminology and language in the manuscript.

Reviewer #2 (Remarks to the Author):

This work shows application of a superhydrophobic coating spray coated on various surfaces (glass, steel and etc.) for signal loss of 5G caused by rain. The coating shows fair durability in different abrasive conditions.

Here are some comments as follows:

1. The prepared SH coating is one of the common coatings with silica nanoparticle treated with an adhesive (for example: Wang, Shuai, Yapeng Li, Xiaoliang Fei, Mingda Sun, Chaoqun Zhang, Yaoxian Li, Qingbiao Yang, and Xia Hong. "Preparation of a durable superhydrophobic membrane by electrospinning poly (vinylidene fluoride)(PVDF) mixed with epoxy-siloxane modified SiO₂ nanoparticles: A possible route to superhydrophobic surfaces with low water sliding angle and high water contact angle." Journal of colloid and interface science 359, no. 2 (2011): 380-388.) The introduction section needs to show how the SH coating presented in this work is different than previously reported coatings.
2. The authors presented a mixture of TEOS and PFDTES that bonded on silica nanospheres in present of an initiator. The PFDTES could also bond on Silica NP without using TEOS layer, could you explain the reason of using the TEOS layer?
3. Figure 4C, is it showing the sliding angle of all condition or just -18°C condition? (It is not clear)
4. According to the manuscript the results of most durability tests show a drop in the sliding angle from almost 1° to 6°-10°. This would result in pinning water droplets during the impact. It would be helpful to show the structure of coating after durability tests, for example SEM images.
- 5- The authors have provided results for the signal loss before and after rainfall. I would suggest that the authors add the results for the repeatability of the measurements and also provide the source of error for the repeatability measurements.
- 6- I would also consider the system's frequency response to be presented before and after rainfall. Also, the repeatability of amplitude and the error sources for the measurement should be provided and mentioned in the paper
- 7- There have been other works on superhydrophobic structures. The authors should consider adding a table and comparing their results with the previously published results.

I have provided some of the recent works in the following. Perhaps the authors can add to this list <https://doi.org/10.1021/acsami.1c07880>, doi: 10.1109/CAMA56352.2022.10002652, doi: 10.1109/APUSNCURSINRSM.2018.8608814

8- Is the proposed solution also effective for icy conditions? As ice accumulation could have an effect on the signal-to-noise ratio, it would be desired that the developed coating could also repel ice.

9- Would the proposed coating also be effective for lower frequencies in the GSM, UMTS or LTE frequency range?

Reviewer #3 (Remarks to the Author):

In this work, the authors reported the preparation of superhydrophobic coatings by spray-coating a suspension of adhesive/fluorinated silica core/shell microspheres onto substrates.

Although the superhydrophobic coatings showed excellent impalement resistance, mechanically robustness and weather resistance, this manuscript lacked enough innovation because some similar component systems were reported. (Advanced Functional Materials 2022, 32, 2113297.; Advanced Functional Materials 2022, 32 (43), 2206014.) Therefore, this manuscript can not meet the high impact requirements of Nature communications. There are some points that may improve the quality of the manuscript.

1. In introduction, the authors need to further highlight the innovation of this work, when compared to their previous reported work (Advanced Functional Materials 2022, 32 (43), 2206014).

2. Peel strength is of vital importance for superhydrophobic coatings in practical applications. Although some measurement such as taber abrasion and tape-peeling were carried out, it would be more persuasive to provide quantitative value of peel strength.

3. The authors demonstrated the practical application of the coatings for preventing rain attenuation of 5G/weather radomes. Dielectric properties of the coatings might influence the performance of 5G/weather radomes. It would be better to evaluate Dielectric properties of the coatings.

4. The relative contents of the various components in superhydrophobic coatings has important effects on their performance. More details on the component systems were suggested to be discussed.

5. The image clarity needs to be improved in Fig. 6.

Responses to Reviewers' Comments

Reviewer #1

This manuscript reports fabrication of superhydrophobic coatings via spray-coating a suspension that consists of silica particles, fluorinated compounds and adhesives. The authors study the mechanical stability of the coatings and also demonstrate application over a radome surface. The coatings exhibit certain level of mechanical robustness due to the hierarchical structures at multiple length scales together with the high thickness of the coating. Overall the novelty of the work is limited for publication in Nature Communications, considering that similar materials and methods have been studied extensively for fabricating superhydrophobic coatings. Please see below additional concerns.

Reply: Thanks for the insightful comment. It seems that we did not show novelty of the work clearly. We have reorganized the novelty as shown below to make it clear, which has also been shown in the title, Introduction, “Design of POA/fluoroPOS@silica coatings” section, and last paragraph of Results and Discussion of the revised manuscript.

First, this is an application-oriented research. So, we can only use simple materials and methods to prepare superhydrophobic coatings with excellent performances. Simple materials and methods are very important for their large-scale preparation and practical applications. In previous studies, the performances of superhydrophobic coatings prepared by such simple materials and methods are not good enough (ACS Appl. Nano Mater. 2020, 3, 5807; Nanoscale 2019, 11, 13853; J. Mater. Chem. A 2015, 3, 13856; J Colloid Interf. Sci. 359, 2011, 380...). In fact, it is very challenging to obtain superhydrophobic coatings with excellent performances via simple materials and methods. Here, based on theoretical analysis, systematic design and optimization (“Design of POA/fluoroPOS@silica coatings” section in the

manuscript), the coatings with excellent impalement resistance (Fig. 2), mechanical robustness (Fig. 3 and Supplementary Table 3) and weather resistance (Fig. 4 and Supplementary Table 5) were prepared using simple materials and methods.

Most of the existing studies about superhydrophobic surfaces manage to enhance certain properties such as mechanical robustness (Nature 2020, 582, 55; Adv. Mater. 2022, 34, 2203792; Adv. Funct. Mater. 2022, 2206014...) or impalement resistance (Adv. Mater. 2018, 30, 1706529...), but seldom give consideration to the comprehensive performances, which is very important for practical applications. In contrast, the impalement resistance, mechanical robustness and weather resistance of the superhydrophobic coatings were balanced in this work, aiming at practical applications.

Furthermore, most of the previous studies only focus on design, preparation and potential applications of superhydrophobic coatings, but lack of large-scale production and practical applications. This work is a precious example realizing design, preparation, large-scale production and practical application of superhydrophobic coatings.

In addition, preventing rain attenuation of 5G/weather radomes is a new application of superhydrophobic coatings. In other word, we report a new application of superhydrophobic coatings and achieve practical application.

1) "With increasing the abrasion or peeling cycles, the newly exposed surface is still the same as the original one". The thickness of the coating plays a key role in the mechanical robustness. The thickness of the coating was not even reported. The mechanical robustness should be studied as a function of the thickness.

Reply: Thanks for the comment. The thickness of the optimal coating is $99.4 \pm 2.3 \mu\text{m}$. The thicknesses of the coatings were recorded using an electronic

digital display micrometer with a resolution of 1 μm (SYA1704569, SYNTEK). The thicknesses of the coatings were calculated according to Eq. S3.

$$\text{Coating thickness} = T_1 - T_2 \quad (\text{S3})$$

where T_1 is the thickness of the substrate with the coating and T_2 is the thickness of the substrate.

The mechanical robustness of the coatings has been studied as a function of thickness via Taber abrasion (Supplementary Fig. 14). With increasing the thickness from 50 μm to 200 μm , the coatings can withstand more Taber abrasion cycles from 30 cycles to 100 cycles. Considering the practical demand for mechanical robustness and the cost of the coatings, the coating with a thickness of ~ 100 μm was selected for further studies.

The data and method have been supplemented in revised manuscript and Supplementary Information.

2) There is significant increase in the sliding angle of the coatings after exposure to different effects (e.g. Figure 4e). These results limit the practical applicability.

Reply: Thanks for the comment. Indeed, the SA of the coating increased significantly to $\sim 30^\circ$ in the long-term weather resistance test in the outdoor environment (Fig. 4e). Note that this happened in a long period of about two years (670 d) rather than a few days. As far as we know, the long-term outdoor stability test of superhydrophobic coatings is rare and the coating in this work is among the best performance (Supplementary Table 5), which in fact paves the way for its practical applications. Of course, for even longer time of practical applications, the long-term outdoor stability of the coating needs to be further improved for example to 5~10 years. This has been mentioned in the last paragraph in the Results and Discussion section in the revised manuscript.

After exposure to other effects such as UV aging, neutral salt spray, immersion in corrosive liquids and treatment at harsh temperature, the SA

remained below 10° (Fig. 4a-c). According to the literature and our experience, $SA < 10^\circ$ represents good superhydrophobicity.

These results prove that the coating has excellent weather resistance, laying a foundation for practical applicability.

3) There are issues regarding the appropriate usage of terminology and language in the manuscript.

Reply: The terminology and language have been carefully checked and modified in the revised manuscript.

Reviewer #2

This work shows application of a superhydrophobic coating spray coated on various surfaces (glass, steel and etc.) for signal loss of 5G caused by rain. The coating shows fair durability in different abrasive conditions. Here are some comments as follows:

1) The prepared SH coating is one of the common coatings with silica nanoparticle treated with and an adhesive (for example: Wang, Shuai, Yapeng Li, Xiaoliang Fei, Mingda Sun, Chaoqun Zhang, Yaoxian Li, Qingbiao Yang, and Xia Hong. "Preparation of a durable superhydrophobic membrane by electrospinning poly (vinylidene fluoride)(PVDF) mixed with epoxy–siloxane modified SiO₂ nanoparticles: A possible route to superhydrophobic surfaces with low water sliding angle and high water contact angle." Journal of colloid and interface science 359, no. 2 (2011): 380-388.)

The introduction section needs to show how the SH coating presented in this work is different than previously reported coatings.

Reply: Thanks for the insightful comment. It seems that we did not show novelty of the work clearly. We have reorganized the novelty as shown below to make it clear, which has also been shown in the title, Introduction, "Design of POA/fluoroPOS@silica coatings" section, and last paragraph of Results and Discussion of the revised manuscript.

First, this is an application-oriented research. So, we can only use simple materials and methods to prepare superhydrophobic coatings with excellent performances. Simple materials and methods are very important for their large-scale preparation and practical applications. In previous studies, the performances of superhydrophobic coatings prepared by such simple materials and methods are not good enough (ACS Appl. Nano Mater. 2020, 3, 5807; Nanoscale 2019, 11, 13853; J. Mater. Chem. A 2015, 3, 13856; J Colloid Interf. Sci. 359, 2011, 380...). For example, in the work mentioned by the reviewer, although the coating was prepared with silica and adhesive, the

mechanical stability of the coating was only evaluated by soaking in a small amount of water (24 h) and did not mention the impalement resistance or weather resistance. In fact, it is very challenging to obtain superhydrophobic coatings with excellent performances via simple materials and methods. Here, based on theoretical analysis, systematic design and optimization ("Design of POA/fluoroPOS@silica coatings" section in the manuscript), the coatings with excellent impalement resistance (Fig. 2), mechanical robustness (Fig. 3 and Supplementary Table 3) and weather resistance (Fig. 4 and Supplementary Table 5) were prepared using simple materials and methods.

Most of the existing studies about superhydrophobic surfaces manage to enhance certain properties such as mechanical robustness (Nature 2020, 582, 55; Adv. Mater. 2022, 34, 2203792; Adv. Funct. Mater. 2022, 2206014...) or impalement resistance (Adv. Mater. 2018, 30, 1706529...), but seldom give consideration to the comprehensive performances, which is very important for practical applications. In contrast, the impalement resistance, mechanical robustness and weather resistance of the superhydrophobic coatings were balanced in this work, aiming at practical applications.

Furthermore, most of the previous studies only focus on design, preparation and potential applications of superhydrophobic coatings, but lack of large-scale production and practical applications. This work is a precious example in the field realizing design, preparation, large-scale production and practical application of superhydrophobic coatings.

In addition, preventing rain attenuation of 5G/weather radomes is a new application of superhydrophobic coatings. In other word, we report a new application of superhydrophobic coatings and achieve practical application.

2) The authors presented a mixture of TEOS and PFDTES that bonded on silica nanospheres in present of an initiator. The PFDTES could also bond on

Silica NP without using TEOS layer, could you explain the reason of using the TEOS layer?

Reply: A small amount of TEOS can enhance superamphiphobicity of the fluoroPOS@silica coating (Supplementary Fig. 1), as TEOS helps hydrolytic condensation of PFDTES on the surface of silica nanoparticles. Obviously, fluoroPOS@silica nanoparticles with higher superamphiphobicity are helpful to improve superhydrophobicity of the POA/fluoroPOS@silica coatings. The data has been supplement in the revised manuscript.

3) Figure 4C, is it showing the sliding angle of all condition or just -18°C condition? (It is not clear)

Reply: Thanks for the advice. Fig. 4c shows the SA of all conditions. As the SA of the coating was 1° during all the four tests in Fig. 4c, the data points overlapped with each other. We have explained this in the revised manuscript.

4) According to the manuscript the results of most durability tests show a drop in the sliding angle from almost 1° to 6°-10°. This would result in pinning water droplets during the impact. It would be helpful to show the structure of coating after durability tests, for example SEM images.

Reply: Thanks for the comment. Although the SA increased from 1° to 6°-10° after most durability tests, we have confirmed that such slight increase in the SA would not result in pinning of water droplets under normal impact conditions like impacting from 1 cm height with a speed of 0.44 m/s or even from 100 cm height with a speed of 4.43 m/s (Supplementary Figs. 10, 11 and Movie 4). In addition, we have supplemented SEM images of the coating after the mechanical robustness test (Supplementary Fig. 13). With increasing the abrasion or peeling cycles, the newly exposed surface is still almost the same as the original one.

5) The authors have provided results for the signal loss before and after rainfall. I would suggest that the authors add the results for the repeatability of the measurements and also provide the source of error for the repeatability measurements.

Reply: We have repeatedly tested the signal loss before and after rainfall for 3 times, and the data have been supplemented in the revised manuscript (Fig. 6 and Supplementary Fig. 22). It is clear that the results have good repeatability. The sources of error for the repeatability measurements are mainly the sensitivity of the test instrument and verticality between the radomes and the signal transmitter & signal receiver.

6) I would also consider the system's frequency response to be presented before and after rainfall. Also, the repeatability of amplitude and the error sources for the measurement should be provided and mentioned in the paper.

Reply: Thanks for the comment. Due to limitation of the instrument, we cannot directly test the system's frequency response and amplitude. We greatly hope the reviewer could understand us on this point. Instead, the signal transmission without radome before and after rainfall was tested for 6 times, which showed good repeatability (Supplementary Fig. 21). This indirectly proves that the system's frequency response and amplitude have good repeatability. The error source is mainly the sensitivity of the test instrument.

7) There have been other works on superhydrophobic structures. The authors should consider adding a table and comparing their results with the previously published results. I have provided some of the recent works in the following. Perhaps the authors can add to this list.

<https://doi.org/10.1021/acsami.1c07880>, doi:
10.1109/CAMA56352.2022.10002652, doi:
10.1109/APUSNCURSINRSM.2018.8608814.

Reply: Thanks for the suggestion. We have compared this work with previous studies in the revised manuscript (Supplementary Tables 2, 5 and 6).

8) Is the proposed solution also effective for icy conditions? As ice accumulation could have an effect on the signal-to-noise ratio, it would be desired that the developed coating could also repel ice.

Reply: According to the suggestion, the anti-icing performance of the coating was supplemented (Supplementary Fig. 16). At -20 °C and 97% relative humidity, the water droplets (60 µL) completely froze after 168.7 ± 4.1 s on the ABS plate, but delayed to 301.7 ± 10.3 s on the POA/fluoroPOS@silica coated ABS plate. Also, the coating can evidently reduce ice adhesion strength from 121 ± 9.3 kPa (the ABS plate) to 39.2 ± 2.8 kPa. Thus, the coating has good anti-icing performance. These data have been supplemented in the revised manuscript.

9) Would the proposed coating also be effective for lower frequencies in the GSM, UMTS or LTE frequency range?

Reply: Thanks for the advice. Due to limitation of the instrument, we cannot verify whether the coating is effective for lower frequencies in the GSM, UMTS or LTE frequency range. However, according to our experience, the coating should be effective, as it can keep the surfaces of radomes dry and clean.

Reviewer #3

In this work, the authors reported the preparation of superhydrophobic coatings by spray-coating a suspension of adhesive/fluorinated silica core/shell microspheres onto substrates. Although the superhydrophobic coatings showed excellent impalement resistance, mechanically robustness and weather resistance, this manuscript lacked enough innovation because some similar component systems were reported. (Advanced Functional Materials 2022, 32, 2113297.; Advanced Functional Materials 2022, 32 (43), 2206014.) Therefore, this manuscript can not meet the high impact requirements of Nature communications. There are some points that may improve the quality of the manuscript.

Reply: Thanks for the insightful comment. It seems that we did not show novelty of the work clearly. We have reorganized the novelty as shown below to make it clear, which has also been shown in the title, Introduction, "Design of POA/fluoroPOS@silica coatings" section, and last paragraph of Results and Discussion of the revised manuscript.

First, this is an application-oriented research. So, we can only use simple materials and methods to prepare superhydrophobic coatings with excellent performances. Simple materials and methods are very important for their large-scale preparation and practical applications. In previous studies, the performances of superhydrophobic coatings prepared by such simple materials and methods are not good enough (ACS Appl. Nano Mater. 2020, 3, 5807; Nanoscale 2019, 11, 13853; J. Mater. Chem. A 2015, 3, 13856; J Colloid Interf. Sci. 359, 2011, 380...). In fact, it is very challenging to obtain superhydrophobic coatings with excellent performances via simple materials and methods. Here, based on theoretical analysis, systematic design and optimization ("Design of POA/fluoroPOS@silica coatings" section in the manuscript), the coatings with excellent impalement resistance (Fig. 2), mechanical robustness (Fig. 3 and Supplementary Table 3) and weather

resistance (Fig. 4 and Supplementary Table 5) were prepared using simple materials and methods.

Most of the existing studies about superhydrophobic surfaces manage to enhance certain properties such as mechanical robustness (Nature 2020, 582, 55; Adv. Mater. 2022, 34, 2203792; Adv. Funct. Mater. 2022, 2206014...) or impalement resistance (Adv. Mater. 2018, 30, 1706529...), but seldom give consideration to the comprehensive performances, which is very important for practical applications. In contrast, the impalement resistance, mechanical robustness and weather resistance of the superhydrophobic coatings were balanced in this work, aiming at practical applications.

Furthermore, most of the previous studies only focus on design, preparation and potential applications of superhydrophobic coatings, but lack of large-scale production and practical applications. This work is a precious example in the field realizing design, preparation, large-scale production and practical application of superhydrophobic coatings.

In addition, preventing rain attenuation of 5G/weather radomes is a new application of superhydrophobic coatings. In other word, we report a new application of superhydrophobic coatings and achieve practical application.

Thus, this work is obviously different from previous studies. For example, there are the following differences between our work and the literature mentioned by the reviewer (Adv. Funct. Mater. 2022, 32, 2113297), which has been added into the revised manuscript.

(i) The coating in the literature was prepared by multi-layer ultrasonic spraying and applied for anti-icing. Multi-layer and ultrasonic spraying are not conducive to large-scale preparation and practical application, especially in the outdoor conditions. Our one-layer design via common spray-coating is much simpler. In fact, one-layer design is preferred for large-area outdoor applications including anti-icing and preventing rain attenuation, as discussed with our customers.

(ii) In the literature, the authors mainly optimized superhydrophobicity and photothermal anti-icing performance of the coating while paying little attention to impalement resistance and weather resistance, etc. Obviously, these are very important for practical applications. In our work, impalement resistance, mechanical robustness and weather resistance of the superhydrophobic coatings were balanced.

(iii) The coatings in the literature is mainly applied on metal surfaces for anti-icing, but in our work on radomes with ABS as the main substrates for preventing rain attenuation, i.e. different substrates and purposes.

(iv) In addition, most of components of the coating in our work are different from those in the literature, except for silica particles but different size and PFDTES but different usage method.

1) In introduction, the authors need to further highlight the innovation of this work, when compared to their previous reported work (Advanced Functional Materials 2022, 32 (43), 2206014).

Reply: Thank you for the advice. Compared with our previous work (Adv. Funct. Mater. 2022, 32, 2206014), the innovation of this work is shown below, which has also been added in the Introduction of the revised manuscript.

In the previous work, we mainly focused on anti-icing performance and mechanical robustness of the coating, but paid little attention to impalement resistance and weather resistance. As a result, we found that the outdoor service life of the coating is not good. Thus, in this work, the impalement resistance, mechanical robustness and weather resistance of the coating were balanced during its design and optimization, forming coatings with excellent comprehensive performance. This work breaks through the bottlenecks of superhydrophobic coatings for long-term practical outdoor applications. This is the main innovation compared with our previous work.

In addition, preventing rain attenuation of 5G/weather radomes is a new application of superhydrophobic coatings. In other word, we report a new application of superhydrophobic coatings and achieve practical application.

Moreover, in the previous work we used ammonia as the catalyst. Thus, the coating suspension is very pungent, which makes it very inconvenient for large-scale spray-coating. In this work, we used sodium methylsilicate as both the reactant and the catalyst in the “Synthesis of fluoroPOS@silica nanoparticles” section to solve the pungent issue in the spray-coating process. This is substantially different from the conventional hydrolytic condensation of silanes with ammonia as the catalyst.

2) Peel strength is of vital importance for superhydrophobic coatings in practical applications. Although some measurement such as Taber abrasion and tape-peeling were carried out, it would be more persuasive to provide quantitative value of peel strength.

Reply: According to the comment, we carried out the peel strength test of the coating. However, due to very low surface energy of the coating, the tape cannot adhere firmly to the coating surface. Therefore, it is impossible to quantitatively test peel strength of the coating. Instead, the hundred-grid adhesion strength test method (ASTM D3359) is widely used to evaluate the adhesion strength of superhydrophobic coatings. The POA/fluoroPOS@silica coating remained intact after the hundred-grid adhesion strength test (Supplementary Fig. 12), which indicates good adhesion strength between the coating and the ABS plate. This has been added in the revised manuscript.

3) The authors demonstrated the practical application of the coatings for preventing rain attenuation of 5G/weather radomes. Dielectric properties of the coatings might influence the performance of 5G/weather radomes. It would be better to evaluate Dielectric properties of the coatings.

Reply: According to the comment, the dielectric strength of the POA/fluoroPOS@silica coating has been tested (38.2 ± 0.88 kV/mm) and added in the revised manuscript.

4) The relative contents of the various components in superhydrophobic coatings has important effects on their performance. More details on the component systems were suggested to be discussed.

Reply: Thanks for the insightful comment. The mass ratio of POA to fluoroPOS@silica nanoparticles has great influences on the coating performance, and thus was supplemented and discussed in the revised manuscript (Supplementary Fig. 3 and Supplementary discussion).

5) The image clarity needs to be improved in Fig. 6.

Reply: The image clarity of Fig. 6 has been improved and a part of it has been moved to Supplementary information.

Reviewer comments second round

Reviewer #1 (Remarks to the Author):

The authors highlighted that this work is application-oriented research in their response letter. My assessment is that the manuscript lacks sufficient scientific novelty for publication in Nature Communications.

From an application perspective: Please note that superhydrophobic coating dispersions are commercially available even at gallon scale (e.g. Never Wet, Rust Oleum). Some of these products have commercial videos on applying such coatings on satellite dishes. In this regard, the presented general application area is well-known. For an application-oriented study, comparison with commercial products and cost analysis are missing.

Overall, the manuscript is a detailed study. Considering the high standards of Nature Communications, the manuscript is NOT recommended for publication.

Reviewer #2 (Remarks to the Author):

My comments have been fairly addressed. I have no further questions or concerns.

Reviewer #3 (Remarks to the Author):

This manuscript has been well revised and could be accepted.

Response to the remaining concerns of Reviewer #1

The authors highlighted that this work is application-oriented research in their response letter. My assessment is that the manuscript lacks sufficient scientific novelty for publication in Nature Communications.

From an application perspective: Please note that superhydrophobic coating dispersions are commercially available even at gallon scale (e.g. Never Wet, Rust Oleum). Some of these products have commercial videos on applying such coatings on satellite dishes. In this regard, the presented general application area is well-known. For an application-oriented study, comparison with commercial products and cost analysis are missing.

Overall, the manuscript is a detailed study. Considering the high standards of Nature Communications, the manuscript is NOT recommended for publication.

Reply: Thanks for the insightful comment. The scientific novelty of this work has been clearly showed in the first round of revision. (1) Although this work is application-oriented research, we have systematically designed the coatings based on theoretical analysis ("Design of POA/fluoroPOS@silica coatings" section in the manuscript), which is the scientific novelty of this work. (2) The impalement resistance, mechanical robustness and weather resistance of the coatings are balanced in this work, which is rare in the literature and is another scientific novelty of this work. (3) This work has simultaneously achieved design, preparation, large-scale production, and practical application of the superhydrophobic coatings in 5G/weather radomes. This work is a good example of superhydrophobic coatings from material design to large-scale production and practical application.

Indeed, there are a few commercial products like Ultra-Ever Dry and NeverWet and videos on applying such coatings on satellite dishes. However, preventing rain attenuation of 5G/weather radomes is still a new application of superhydrophobic coatings. In addition, there are numerous videos on the web about superhydrophobic coatings, which do not really mean practical applications of the coatings but in many cases just show potential applications. Moreover, the performance of the coating outperforms the commercial products in terms of superhydrophobicity, mechanical stability and simplicity as supplemented in the revised manuscript (Supplementary Table 6). The cost of the coating is ~17 USD per liter and this has been supplemented.